# Being Underweight Is Associated with Increased Risk of Sudden Cardiac Death in People with Diabetes Mellitus

**DOI:** 10.3390/jcm12031045

**Published:** 2023-01-29

**Authors:** Yun Gi Kim, Kyung-Do Han, Seung-Young Roh, Joo Hee Jeong, Yun Young Choi, Kyongjin Min, Jaemin Shim, Jong-Il Choi, Young-Hoon Kim

**Affiliations:** 1Division of Cardiology, Department of Internal Medicine, Korea University College of Medicine, Korea University Anam Hospital, Seoul 02841, Republic of Korea; 2Department of Statistics and Actuarial Science, Soongsil University, Seoul 06978, Republic of Korea; 3Division of Cardiology, Department of Internal Medicine, Korea University College of Medicine, Korea University Guro Hospital, Seoul 08308, Republic of Korea; 4Division of Cardiology, Department of Internal Medicine, Sanggye Paik Hospital, Inje University College of Medicine, Seoul 01757, Republic of Korea

**Keywords:** underweight, diabetes mellitus, sudden cardiac death

## Abstract

Background: Diabetes mellitus (DM) can cause various atherosclerotic cardiovascular disease including sudden cardiac death (SCD). The impact of being underweight on the risk of SCD in people with DM remains to be revealed. We aimed to evaluate the risk of SCD according to body-mass index (BMI; kg/m^2^) level in DM population. Methods: We used a nationwide healthcare insurance database to conduct this study. We identified people with DM among those who underwent nationwide health screening during 2009 to 2012. Medical follow-up data was available until December 2018. Results: A total of 2,602,577 people with DM with a 17,851,797 person*year follow-up were analyzed. The underweight group (BMI < 18.5) showed 2.4-fold increased risk of SCD during follow-up (adjusted-hazard ratio [HR] = 2.40; 95% confidence interval [CI] = 2.26–2.56; *p* < 0.001). When normal-BMI group (18.5 ≤ BMI < 23) was set as a reference, underweight group (adjusted-HR = 2.01; 95% CI = 1.88–2.14) showed even higher risk of SCD compared with the obesity group (BMI ≥ 30; adjusted-HR = 0.89; 95% CI = 0.84–0.94). When BMI was stratified by one unit, BMI and SCD risk showed a U-curve association with the highest risk observed at low BMI levels. The lowest risk was observed in 27 ≤ BMI < 28 group. The association between being underweight and increased SCD risk in DM people was maintained throughout various subgroups. Conclusions: Being underweight is significantly associated with an increased risk of SCD in the DM population. A steep rise in the risk of SCD was observed as the BMI level decreased below 23. The lowest risk of SCD was observed in 27 ≤ BMI < 28 group.

## 1. Introduction

Significant social and economic losses occur every day due to sudden cardiac death (SCD) [1,2]. Despite substantial improvements in the overall management of SCD victims, survival and especially neurologically intact survival after SCD is still not satisfactory [3,4,5,6,7]. An intrinsic obstacle for SCD management is the narrow window of timely intervention and adequate cardiopulmonary resuscitation immediately after an SCD event, which is critical for survival of victims [3,4,5,8,9].

Diabetes mellitus (DM) is a known risk factor for SCD and prior studies demonstrated a robust association between fasting blood glucose and risk of SCD [10,11,12]. A recent study based on the Korean nationwide healthcare database revealed an 80% increased risk of SCD in DM patients, and even higher risk (three-fold) in uncontrolled DM patients, despite improvements in oral antidiabetic medications and coronary revascularization capabilities [12]. Despite the strong association between DM and SCD, identification of specific subgroups such as uncontrolled DM patients who are subject to substantially increased risk of SCD will be essential for primary prevention of SCD in DM patients.

Prior studies reported that obesity is a significant predictor for occurrence of SCD in the general population (including those with and without diabetes) [13,14,15,16]. However, the association between being underweight and SCD is not fully understood. Jee et al. revealed that high body-mass index (BMI) is associated with a significantly increased risk of cardiovascular death in the general population of South Korea. However, low BMI did not show any significant association with cardiovascular death [14]. Another study conducted with 2.3 million of the general population of Israel with 40 years of follow-up also reported a considerably increased risk of SCD in obese people but not in people who are underweight [15]. In contrast, Chang et al. reported in their meta-analysis that there was a 59% increased risk of all-cause mortality in diabetic people who were underweight [17].

Both under- and overweight conditions are important risk factors for DM and adequate weight control is recommended for DM patients to prevent cardiovascular complications [18]. In this study, we aimed to evaluate the risk of SCD in DM patients who are underweight. Due to the low incidence of SCD, we utilized nationwide health screening data to analyze a sufficient number of SCD events and to enable various subgroup analyses.

## 2. Patients and Methods

### 2.1. Nationwide Database

This study was based on the Korean National Health Insurance Service (K-NHIS) database. All citizens of South Korea are mandatory subscribers of the K-NHIS and medical data stored in the K-NHIS can represent the entire people of South Korea. Claims of various diagnostic codes based on International Classification of Disease, tenth edition (ICD-10) such as hypertension, DM, dyslipidemia, atrial fibrillation, or heart failure, and a prescription history of various drugs are recorded in the K-NHIS database. The K-NHIS offers a biennial health screening program to its subscribers who are ≥20 years old. Body weight and height are directly measured during health screening, enabling classification of participants according to BMI value. Additional medical data such as blood pressure, waist circumference, amount of alcohol consumption, smoking status, physical activity level, various laboratory tests such as complete blood cell count, liver function, renal function, fasting blood glucose (FBG), and lipid profile are measured during health screening.

### 2.2. Participants

Among people who underwent nationwide health screening during 2009 to 2012, those with a prior diagnosis of DM were enrolled. Only people with type 2 DM was screened and people with type 1 DM was not included in this study. Those with prior diagnosis of SCD and age < 20 years at health screening were excluded from the study. Baseline medical history such as hypertension, DM, or prior SCD event was identified by the data obtained during January 2002 to December 2008. Medical data gathered until December 2018 were analyzed. The nature of subscription to the K-NHIS is mandatory and exclusive and therefore, no follow-up losses exist except for migration.

The Institutional Review Board of Korea University Medicine Anam Hospital and the official review committee of the K-NHIS approved this study. Written informed consent was waived considering the retrospective nature of the study. The ethical guidelines of the 2013 Declaration of Helsinki and legal regulations of South Korea were strictly observed throughout the study.

### 2.3. Sudden Cardiac Death

The main outcome of this study was the occurrence of SCD. In this study, we intended to identify death events that occurred unexpectedly and suddenly which are assumed to originate from cardiac problems. We first identified sudden death events with ICD-10 codes. Subsequently, we excluded potential non-cardiac sudden death events to exclusively identify SCD among sudden death events. Aborted SCD events were also included as SCD. Performance of cardiopulmonary resuscitation without ICD-10 codes for SCD was also classified as SCD events.

The identification of occurrence of an SCD event was based on the ICD-10 codes claimed from emergency departments throughout South Korea. Only claims accompanied by cardiopulmonary resuscitation or a declaration of death were considered as SCD. In-hospital claim of ICD-10 codes for SCD was not included in this analysis. The ICD-10 codes used to identify SCD were as follows: I46.0 (cardiac arrest with successful resuscitation), I46.1 (sudden cardiac arrest), I46.9 (cardiac arrest, cause unspecified), I49.0 (ventricular fibrillation and flutter), R96.0 (instantaneous death), and R96.1 (death occurring less than 24 h from symptom onset). Both aborted and non-aborted SCD were included as a main outcome.

The incidence of SCD was defined as the number of events per 1000 person*year follow-up. Due to ICD-10 coding-based detection of main outcomes, the claims for SCD or death that occurred within one year after health screening were not counted as a main outcome. For example, claims of SCD codes immediately after health screening can be actual SCD events after health screening or just repeat claims of SCD which happened before health screening. The robustness of our coding strategy for SCD and other medical conditions was validated in prior studies [12,19,20,21,22,23,24,25].

If the participants had a prior diagnosis of asphyxia, suffocation, drowning, gastrointestinal bleeding, cerebral hemorrhage, ischemic stroke, sepsis, anaphylaxis, trauma, lightning strike, electric shock, or burn within 6 months (including those claimed simultaneously with ICD-10 codes for SCD) of the diagnosis of SCD, the event was not counted as a SCD event.

### 2.4. Definitions

Underweight was defined as BMI < 18.5 kg/m^2^. Normal body weight, high-normal body weight, pre-obesity, and obesity were defined as 18.5 ≤ BMI < 23.0, 23.0 ≤ BMI < 25.0, 25.0 ≤ BMI < 30.0, and BMI ≥ 30.0, respectively.

Either FBG (FBG ≥ 126 mg/dL for DM and FBG 100–125 mg/dL for impaired fasting glucose (IFG)) or a claim of ICD-10 codes for DM or IFG by a physician was used for the diagnosis of DM or IFG. Hypertension was diagnosed based on ICD-10 codes for hypertension or measured blood pressure during nationwide health screening. The Modification of Diet in Renal Disease equation was used to define chronic kidney disease (estimated glomerular filtration rate < 60 mL/min/1.73 m^2^). A self-questionnaire acquired during health screening was used to define smoking status (current-smoker: ≥100 cigarettes in their lifetime; ex-smoker: ≥100 cigarettes in their lifetime, but had not smoked within one month of health screening; never-smoker: <100 cigarettes in their lifetime) and alcohol consumption (nondrinker: 0 g of alcohol per week; mild- to moderate-drinker: <210 g of alcohol per week; heavy-drinker, ≥210 g of alcohol per week). Regular physical activity was defined as having one or more sessions of high (such as running, climbing, or intense bicycle activities) or moderate physical activity (such as walking fast, tennis, or moderate bicycle activities) in a week.

### 2.5. Statistical Analysis

The Student’s *t*-test was used to compare continuous variables which were expressed as mean ± standard deviation. Categorical variables were compared using the chi-square test or Fisher’s exact test, as appropriate. Cox-regression analysis was used to calculate raw and adjusted hazard ratios with 95% confidence intervals (CIs). People with missing data were excluded from the study and no imputation was done. Statistical significance was defined as *p* values ≤ 0.05 in two-tailed tests. All statistical analyses were performed with SAS version 9.2 (SAS Institute, Cary, NC, USA).

### 2.6. Data Availability Statement

The data underlying this article are available in the article and in its online Appendix A.

## 3. Results

### 3.1. Study Population

We identified a total of 2,746,079 people with DM who underwent nationwide health screening during 2009 to 2012. People were excluded from the analysis if they were under 20 years (*n* = 390), had missing data (117,446), had a prior diagnosis of SCD during the screening period (2002 to 2008; *n* = 934), or had a death or SCD event within one year after health screening (*n* = 24,732). People were followed until December 2018 with a total of 17,851,797 person*year follow-up duration. Mean follow-up period per person was 6.86 years and a total of 26,341 SCD events were detected. The flow of this study is summarized in Figure 1.

Baseline demographics according to BMI value are described in Table 1. A significant difference was observed throughout various parameters such as age, sex, smoking status, alcohol consumption, income level, and prevalence of various medical diseases such as hypertension or dyslipidemia. People who were underweight (BMI < 18.5) were significantly older (59.7 years vs. 57.4 years; *p* < 0.001); were more likely to be women (42.9% vs. 39.9%; *p* < 0.001), current smokers (34.0% vs. 25.7%; *p* < 0.001), non-drinkers (63.7% vs. 57.2%; *p* < 0.001), and lowest income quartile (25.0% vs. 20.9%; *p* < 0.001); were less likely to have regular exercise (14.6% vs. 20.7%; *p* < 0.001); had lower prevalence of hypertension (38.4% vs. 57.1%; *p* < 0.001) and dyslipidemia (21.9% vs. 42.3%; *p* < 0.001). The prevalence of DM for five or more years (30.8% vs. 31.1%; *p* = 0.156) and prescription of three or more oral antidiabetic medications (14.8% vs. 14.5%; *p* = 0.112) were similar between underweight and non-underweight groups. However, the percentage of people on insulin therapy was significantly higher in the underweight group (14.4% vs. 8.7%; *p* < 0.001).

### 3.2. Risk of Sudden Cardiac Death

The raw incidence of SCD was significantly higher in people who were underweight (4.38 vs. 1.43; HR = 3.11; 95% CI = 2.93–3.31; *p* < 0.001; Table 2). Kaplan–Meier curve analysis revealed a steady and continuous divergence of cumulative events of SCD between underweight and non-underweight group (log rank *p* < 0.001; Appendix A). After adjustment of confounders (age, sex, income level, smoking history, alcohol consumption, regular physical activity, hypertension, dyslipidemia, fasting glucose, duration of DM, use of insulin, and number of oral antidiabetic medications), people who were underweight showed a 2.4-fold increased risk of SCD during follow-up (adjusted-HR = 2.40; 95% CI = 2.26–2.56; *p* < 0.001; Table 2). The risk of SCD differed across BMI value and with normal body weight (18.5 ≤ BMI < 23) as a reference, pre-obesity (25 ≤ BMI < 30; adjusted-HR 0.71; 95% CI = 0.69–0.74; *p* < 0.001) and being underweight (adjusted-HR = 2.01; 95% CI = 1.88–2.14; *p* < 0.001) had the lowest and highest risk, respectively (Table 2). When people were classified by one unit of BMI (kg/m^2^), people with 27 ≤ BMI < 28 had the lowest risk of SCD and people with BMI < 17 experienced the highest risk (Figure 2 and Appendix A).

### 3.3. Subgroup Analysis

The impact of being underweight was analyzed in various subgroups. Being underweight was associated with significantly increased risk of SCD regardless of age, sex, income level, smoking history, alcohol consumption, regular physical activity, hypertension, dyslipidemia, fasting glucose, duration of DM, use of insulin, and number of oral antidiabetic medications (Figure 3 and Appendix A). However, significant interactions were observed with age, alcohol consumption, presence of dyslipidemia, and insulin use. People in the age range 40 ≤ age < 64 (adjusted-HR = 3.36; 95% CI = 3.06–3.70; *p* for interaction < 0.001), who were heavy-drinkers (adjusted-HR = 3.19; 95% CI = 2.67–3.82; *p* for interaction = 0.010), those without dyslipidemia (adjusted-HR = 2.50; 95% CI = 2.33–2.68; *p* for interaction = 0.015), and who were not on insulin therapy (adjusted-HR = 2.49; 95% CI = 2.32–2.67; *p* for interaction = 0.031) showed a stronger association between being underweight and SCD compared with their counterparts.

## 4. Discussion

The current study demonstrated that (i) being underweight was associated with a 2.4-fold increased risk of SCD in DM patients; (ii) DM patients who were pre-obese (25 ≤ BMI < 30), and more specifically, 27 ≤ BMI < 28 showed the lowest risk of SCD; (iii) a stronger association between being underweight and the risk of SCD was observed in middle-aged people (40–65), heavy-drinkers, and those without dyslipidemia and insulin therapy. The strength of this study is its large sample size (*n* = 2,602,577), sufficient SCD events (*n* = 26,341), and long follow-up duration (mean 6.86 years per person) which enabled various subgroup analyses.

Diabetes mellitus is associated with various cardiovascular complications including SCD. Since SCD is associated with a low chance of survival and even lower chance of neurologically intact survival, prevention rather than treatment might be the appropriate strategy to reduce the socioeconomic burden [26,27,28,29]. The observed association between being underweight and SCD in DM patients indicates a specific subset of patients that might benefit from intensified monitoring and medical treatment for primary prevention of SCD.

### 4.1. Obesity and SCD

The cluster of glucose intolerance, central obesity, dyslipidemia, and elevated blood pressure is the hallmark of metabolic syndrome [30]. Our recent study revealed a significantly increased risk of SCD in people with metabolic syndrome which was in accordance with prior studies [25,31,32]. However, the current study revealed different results with DM people at the pre-obesity stage having the lowest risk of SCD. Obese DM patients also showed a 11% lower risk of SCD compared with normal weight DM patients in this study. The potential cause for a lack of association (or even inverse association) between obesity and SCD in DM population is not fully explained. However, the degree of obesity and ethnic difference can be important. When participants were stratified by one unit of BMI, people with BMI ≥ 35 showed a clear increase in the risk of SCD which suggests that extreme obesity can increase the risk of SCD. People with BMI ≥ 35 only comprised 0.89% of the entire cohort which is much less than in western populations. This ethnic difference might have diluted the association between obesity and SCD.

### 4.2. Being Underweight and SCD

Unlike obesity, being underweight was clearly associated with increased risk of SCD. A 2.4-fold increased risk of SCD after adjustment of confounders, steady divergence observed in Kaplan–Meier curve analysis, and linear increase in adjusted-HR per one unit decline in BMI within the underweight group all suggest a strong and robust association between being underweight and SCD in people with DM. Although the results of this study can only suggest an association and not causality, intensified monitoring of heart function or coronary artery disease might be justified in DM patients who are underweight, considering a significantly increased risk in comparison with non-underweight people and also the observed absolute incidence (4.38 SCD events per 1000 person*year).

The underlying pathophysiology of this association is not clear. One possible explanation is the genetic susceptibility of underweight DM patients. Those who develop DM despite being underweight might have been more genetically susceptible [33]. A study by Perry et al. revealed that a variant (rs8090011) in the *LAMA1* gene was associated with DM in the subset of underweight cases [33]. Several genetic variations leading to DM in underweight populations might exist and it could be possible that these genetic variations might make a substantial contribution to occurrence of SCD. Whether genetic characteristics unique to underweight DM patients can impact SCD risk is an area of future research. Another possible explanation is the severity of DM. The severity of DM might be more severe in underweight DM patients. However, the percentage of DM duration ≥ 5 years and prescription of ≥3 oral antidiabetic medications were similar between underweight vs. non-underweight groups. Although the prescription rate of insulin was significantly higher in the underweight group (14.4% vs. 8.7%), the difference was adjusted in our multivariate model. Despite significant interaction existed according to insulin use with those without insulin having higher adjusted-HR, the underweight group had a significantly higher risk of SCD in both without insulin and on insulin groups. Presence of undetected confounders might also explain the obesity paradox observed in this study. Although we adjusted various covariates such as age, sex, income, alcohol, smoking, physical activity, hypertension, dyslipidemia, and duration and treatment of DM, there can still be unmeasured covariates such as ongoing inflammation. People might have been already ill at the time of BMI measurement and this issue can contribute to obesity paradox. However, we solved this problem by excluding any SCD events or death within one year after nationwide health screening.

### 4.3. Limitations

Several limitations exist in this study. Despite validation of our coding strategy in various prior publications, coding inaccuracies can be problematic due to the retrospective nature of current study [19,23,24]. Generalizability is another limitation since this analysis was exclusively based on an East Asian population. We were not able to measure serial BMI value which limited additional analysis regarding the impact of temporal change of BMI on the risk of SCD. The observed higher risk of SCA in underweight people can be due to unmeasured confounders. Although we excluded potential non-cardiac causes of sudden death, some of them might have been included in primary events. Since this study was based on claim data of ICD-10 codes, we were not able to clarify specific cardiac causes of SCD. Finally, hemoglobin A1c level was not available although we had fasting blood glucose data.

## 5. Conclusions

Compared with normal weight, being underweight was significantly associated with increased risk of SCD in people with DM. In contrast, high normal weight, pre-obesity, and obesity was associated with a decreased risk of SCD in DM population. The BMI value associated with the lowest risk of SCD was 27 kg/m^2^. Our study raises awareness of SCD risk in DM patients with accompanying underweight. Whether correction of underweight can decrease the risk of SCD remains to be explored.

## Figures and Tables

**Figure 1 jcm-12-01045-f001:**
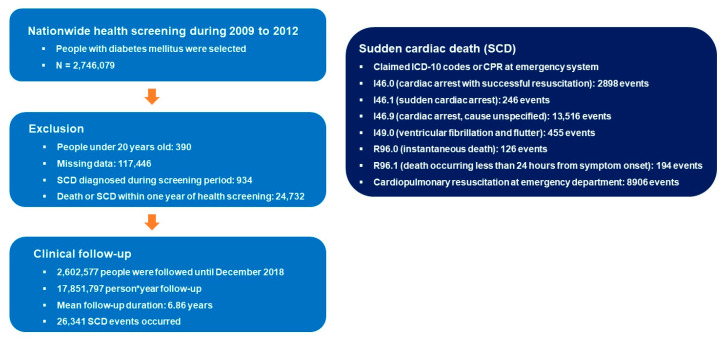
Flow of the study. ICD-10: International Classification of Disease, tenth edition; SCD: sudden cardiac death.

**Figure 2 jcm-12-01045-f002:**
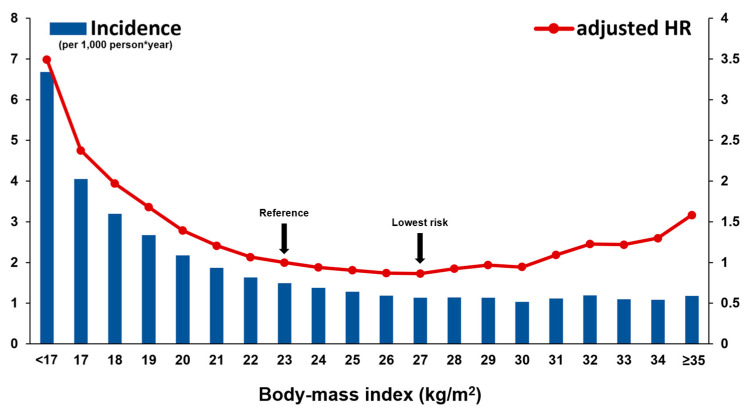
Risk of SCD according to BMI value. The raw incidence of SCD was higher in diabetic patients with low BMI. In the multivariate model, the risk of SCD and BMI showed a U-curve association; the risk was significantly higher in patients with low BMI than high BMI. The multivariate model is adjusted for age, sex, income level, smoking history, alcohol consumption, regular physical activity, hypertension, dyslipidemia, fasting glucose, duration of DM, use of insulin, and number of oral antidiabetic medications. HR: hazard ratio; SCD: sudden cardiac death.

**Figure 3 jcm-12-01045-f003:**
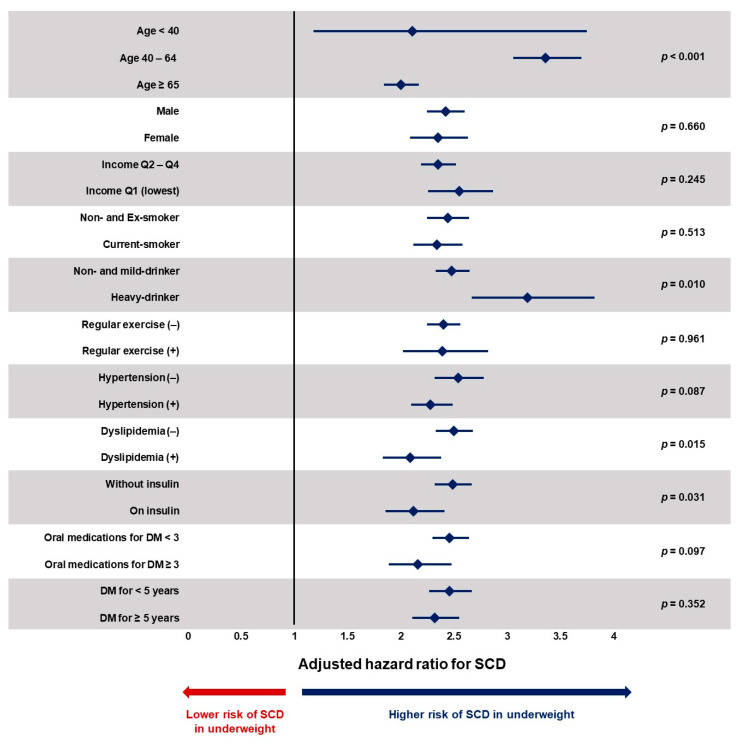
Subgroup analysis. DM: diabetes mellitus; SCD: sudden cardiac death. Despite significant interactions with age, alcohol consumption, dyslipidemia, and insulin use, being underweight was associated with a significantly increased risk of SCD in all subgroups. Underweight is defined as those with BMI less than 18.5 kg/m^2^. Values are expressed as adjusted hazard ratio with 95% confidence interval. Hazard ratios are adjusted for age, sex, income level, smoking history, alcohol consumption, regular physical activity, hypertension, dyslipidemia, fasting glucose, duration of DM, use of insulin, and number of oral antidiabetic medications.

**Table 1 jcm-12-01045-t001:** Baseline characteristics of people according to BMI value.

	Body-Mass Index (BMI)							
	BMI < 18.5	18.5 ≤ BMI < 23	23 ≤ BMI < 25	25 ≤ BMI < 30	BMI ≥ 30	*p* Value	BMI ≥ 18.5	*p* Value (<18.5 vs. ≥18.5)
** *n* **	41,598	648,206	645,444	1,066,439	200,890		2,560,979	
**Age, years**	59.7 ± 16.1	58.6 ± 12.9	58.3 ± 11.7	57.0 ± 11.8	53.0 ± 13.0	<0.001	57.4 ± 12.3	<0.001
**Age groups**						<0.001		<0.001
<40	5172 (12.4%)	47,580 (7.3%)	35,757 (5.5%)	76,107 (7.1%)	32,037 (16.0%)		191,481 (7.5%)	
40 ≤ age < 65	18,663 (44.9%)	378,603 (58.4%)	406,130 (62.9%)	690,485 (64.8%)	126,645 (63.0%)		1,601,863 (62.6%)	
≥65	17,763 (42.7%)	222,023 (34.3%)	203,557 (31.5%)	299,847 (28.1%)	42,208 (21.0%)		767,635 (30.0%)	
**Sex**						<0.001		<0.001
Male	23,765 (57.1%)	382,404 (59.0%)	403,135 (62.5%)	653,020 (61.2%)	99,974 (49.8%)		1,538,533 (60.1%)	
Female	17,833 (42.9%)	265,802 (41.0%)	242,309 (37.5%)	413,419 (38.8%)	100,916 (50.2%)		1,022,446 (39.9%)	
**Income (lowest quartile)**	10,377 (25.0%)	142,184 (21.9%)	132,778 (20.6%)	217,179 (20.4%)	43,936 (21.9%)	<0.001	536,077 (20.9%)	<0.001
**Smoking**						<0.001		<0.001
Non	22,586 (54.3%)	360,640 (55.6%)	352,969 (54.7%)	589,443 (55.3%)	122,660 (61.1%)		1,425,712 (55.7%)	
Ex	4865 (11.7%)	103,684 (16.0%)	126,303 (19.6%)	215,921 (20.3%)	30,257 (15.1%)		476,165 (18.6%)	
Current	14,147 (34.0%)	183,882 (28.4%)	166,172 (25.8%)	261,075 (24.5%)	47,973 (23.9%)		659,102 (25.7%)	
**Drinking**						<0.001		<0.001
Non	26,504 (63.7%)	383,752 (59.2%)	365,192 (56.6%)	595,316 (55.8%)	120,730 (60.1%)		1,464,990 (57.2%)	
Mild	11,386 (27.4%)	205,557 (31.7%)	217,857 (33.8%)	357,372 (33.5%)	59,925 (29.8%)		840,711 (32.8%)	
Heavy	3708 (8.9%)	58,897 (9.1%)	62,395 (9.7%)	113,751 (10.7%)	20,235 (10.1%)		255,278 (10.0%)	
**Regular exercise**	6065 (14.6%)	134,781 (20.8%)	142,392 (22.1%)	219,627 (20.6%)	33,160 (16.5%)	<0.001	529,960 (20.7%)	<0.001
**Hypertension**	15,965 (38.4%)	301,988 (46.6%)	352,520 (54.6%)	664,559 (62.3%)	143,015 (71.2%)	<0.001	1,462,082 (57.1%)	<0.001
**Dyslipidemia**	9092 (21.9%)	223,499 (34.5%)	268,170 (41.6%)	490,648 (46.0%)	99,901 (49.7%)	<0.001	1,082,218 (42.3%)	<0.001
**DM duration, ≥5 years**	12,806 (30.8%)	227,952 (35.2%)	214,249 (33.2%)	307,177 (28.8%)	47,331 (23.6%)	<0.001	796,709 (31.1%)	0.156
**On Insulin**	5983 (14.4%)	68,633 (10.6%)	55,762 (8.6%)	82,799 (7.8%)	14,764 (7.4%)	<0.001	221,958 (8.7%)	<0.001
**Oran antidiabetics, ≥3**	6144 (14.8%)	102,545 (15.8%)	95,288 (14.8%)	146,073 (13.7%)	27,257 (13.6%)	<0.001	371,163 (14.5%)	0.112
**BMI (kg/m^2^)**	17.4 ± 0.9	21.4 ± 1.2	24.0 ± 0.6	26.9 ± 1.3	32.3 ± 2.4	<0.001	25.2 ± 3.3	<0.001
**Waist circumference (cm)**	70.0 ± 6.4	78.0 ± 6.0	83.5 ± 5.4	89.2 ± 6.1	98.7 ± 7.8	<0.001	85.7 ± 8.5	<0.001
**Systolic BP (mmHg)**	122.3 ± 17.4	126.1 ± 16.1	128.5 ± 15.5	130.6 ± 15.3	133.7 ± 15.9	<0.001	129.2 ± 15.8	<0.001
**Diastolic BP (mmHg)**	75.3 ± 10.8	76.9 ± 10.2	78.5 ± 10.0	80.2 ± 10.1	82.6 ± 10.7	<0.001	79.1 ± 10.3	<0.001
**Fasting glucose (mg/dL)**	152.1 ± 65.5	146.4 ± 51.7	144.3 ± 46.4	143.6 ± 43.8	145.7 ± 45.2	<0.001	144.7 ± 46.7	<0.001
**Total cholesterol (mg/dL)**	183.0 ± 41.7	191.6 ± 41.9	196.2 ± 42.6	199.1 ± 43.0	202.6 ± 43.4	<0.001	196.7 ± 42.8	<0.001
**HDL (mg/dL)**	59.1 ± 28.5	54.6 ± 25.5	51.9 ± 23.5	50.6 ± 22.9	50.2 ± 22.0	<0.001	51.9 ± 23.7	<0.001
**LDL (mg/dL)**	101.4 ± 41.5	108.8 ± 40.2	111.4 ± 40.8	112.2 ± 41.6	114.0 ± 41.9	<0.001	111.3 ± 41.1	<0.001
**Triglyceride (mg/dL)**	101.5 (100.9–102.0)	123.6 (123.5–123.8)	144.8 (144.6–145.0)	160.7 (160.6–160.9)	171.4 (171.0–171.8)	<0.001	147.2 (147.1–147.3)	<0.001

BMI: body-mass index; BP: blood pressure; DM: diabetes mellitus; HDL: high-density lipoprotein; LDL: low-density lipoprotein.

**Table 2 jcm-12-01045-t002:** Impact of underweight on SCD in DM patients.

	*N*	SCD	Follow-Up Duration (Person-Years)	Incidence	Hazard Ratio with 95% Confidence Interval		
					Non-Adjusted	Model 1	Model 2
**Underweight**							
No (BMI ≥ 18.5)	2,560,979	25,259	17,604,724	1.43	1 (reference)	1 (reference)	1 (reference)
Yes (BMI < 18.5)	41,598	1082	247,073	4.38	3.11 (2.93–3.31)	2.56 (2.41–2.72)	2.40 (2.26–2.56)
**BMI value**							
BMI < 18.5	41,598	1082	247,073	4.38	2.29 (2.15–2.44)	2.01 (1.94–2.20)	2.01 (1.88–2.14)
18.5 ≤ BMI < 23	648,206	8491	4,355,959	1.95	1 (reference)	1 (reference)	1 (reference)
23 ≤ BMI < 25	645,444	6401	4,457,801	1.44	0.73 (0.71–0.76)	0.75 (0.73–0.78)	0.77 (0.75–0.80)
25 ≤ BMI < 30	1,066,439	8844	7,408,578	1.19	0.61 (0.59–0.63)	0.69 (0.67–0.71)	0.71 (0.69–0.74)
BMI ≥ 30	200,890	1523	1,382,385	1.10	0.57 (0.54–0.60)	0.89 (0.84–0.94)	0.89 (0.84–0.94)

Incidence is per 1000 person-years of follow-up. BMI: body-mass index; DM: diabetes mellitus; SCD: sudden cardiac death. Model 1: adjusted for age and sex. Model 2: adjusted for age, sex, income level, smoking history, alcohol consumption, regular physical activity, hypertension, dyslipidemia, fasting glucose, duration of DM, use of insulin, and number of oral antidiabetic medications.

## Data Availability

The raw data underlying this article cannot be shared publicly due to privacy reasons and legal regulations of the Republic of Korea. The raw data is stored and analyzed only in the designated server managed by the K-NHIS

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
