# Peer review of "Being Underweight Is Associated with Increased Risk of Sudden Cardiac Death in People with Diabetes Mellitus"

_jcm, 2023, doi:10.3390/jcm12031045_

Round 1
Reviewer 1 Report
This hugh Korean population data study deals with the SCD risk in patients with diabetes mellitus. Specifically correlation between BMI and SCD was studied and showed convincingly the highest risk in patients with subnormal BMI. In patients with BMI below 18.5 the adjusted risk was 2.4 x the risk in patients with higher BMI. Also the risk in underweight patients appeared to be higher than in obese patients. The study is original, well described with appropriate confounder analysis and limitations section. Clinically relevant.
Minor specific comments and corrections:
-lines 58-64. The studies in references 10-13 are in non-diabetic patients. This should be more clearly indicated since this part of text is embedded in between studies in diabetic patients.
-lines 104-107. A table should be added in the results session or as supplement indicating how often diagnosis of SCD according to the different ICD-10 codes.
-line 107. Use of Code R96.1 may be criticized and needs explanation.
-In table 1 the number n= 2560,797 is different from numbers in figure 1. Explanation needed.
-line 273. A should be a.
Author Response
Dear reviewer,
We appreciate your time and effort to provide insightful comments and suggestions that are kindly aimed to improve the quality of our manuscript.
Since our response to your comments contains images, we uploaded our letter as a word file. Thank you.
Sincerely yours,
Jong-Il Choi, MD, PhD, MHS.

Reviewer 2 Report
Kim et al. present this retrospective analysis from a large South Korean registry, which again identifies an increased mortality risk for patients with low BMI. Their available sample size is vast and their results are of importance and of high impact. I have the following concerns, which should be addressed by the authors:
- In their Introduction section and also later in the manuscript, the authors refer to another similar manuscript under revision in another journal (Sci. Rep). Such is unusual to my experience and not scientifically sound, as they now have two manuscripts on peer review process in different journals, likely based on pretty much the same subject with the same data material and methods and citing each other. A manucript in revision is by no means accepted (I have had the painful experience of having been rejected once after revision). And why did the authors decide to separate their findings in the two manuscripts?
- At section "2.3 Sudden cardiac death", the authors state they also included patients with ICD-10 code R96.1 as SCD. The description of this code includes deaths which were not sudden, but unexplained and occurring within 24h after onset of symptoms. That is not in line with my understanding of SCD. If the authors maintain their position into including such deaths as SCD, then they have to elaborate on their reasons to do so and cite accordingly.
- Also on section 2.3, patients state they excluded SCDs when several other events such as asphyxia, soffocation, drowning etc. priorly occurred in the last 6 months. Which is fine, but what about patients when such events occurred simultaneuously with SCD? PLease clarify.
- SCD is not universally defined, pressumed to occur due to ventricular fibrillation or other deadly arrhythmias. Yet such is difficult to be evidenced in practice and thus SCD is also applied to sudden deaths without eidence of such cardiac events. THe authors of this manuscript should better describe their used/chosen definition of SCD, so that a better understanding of which kind of deaths were are talking about here is cleaer.
- The authors include some other concommitant diagnosis in their analysis, but ommit information on other cardiac diagnosis at the time of death, such as myocardial infarction, cardiogenic shock or pulmonary embolism. It is possible that a SCD diagnosis as well as a cardiac diagnosis were simultaneously documented on these patients. That information should be included in the manuscript, analyzed and properly discussed.
Author Response
Dear reviewers,
We appreciate your time and effort to provide insightful comments and suggestions that are kindly aimed to improve the quality of our manuscript.
Since our response to your comments contains images, we uploaded our letter as a word file. Thank you.
Sincerely yours,
Jong-Il Choi, MD, PhD, MHS.

Round 2
Reviewer 2 Report
I have no further suggestions for the authors.